# Development and validation of a questionnaire assessing household work limitations (HOWL-Q) in women with rheumatoid arthritis

**Ana Belén Ortiz-Haro[1]⊙, Abel Lerma-Talamantes[2]⊙, Ángel Cabrera-Vanegas[1]⊙, Irazú Contreras-Yáñez[1]⊙, Virginia Pascual-Ramos⊙[1]⊙ \***

1 Department of Immunology and Rheumatology, Instituto Nacional de Ciencias Médicas y Nutrición Salvador Zubirán, Mexico City, Mexico, 2 Health Sciences Institute, Psychology Academic Area, Universidad Autónoma del Estado de Hidalgo, Pachuca, Mexico

⊙ These authors contributed equally to this work.

\* virtichu@gmail.com

**Data Availability Statement:** All relevant data are within the manuscript and its Supporting Information files.

## Abstract

### Introduction

Rheumatoid arthritis (RA) has female preponderance and interferes with the ability to perform job roles. Household work has 2 dimensions, paid and unpaid. There is not a validated instrument that assesses the impact of RA on limitations to perform household work. We report the development and validation of a questionnaire that assesses such limitations, the HOWL-Q.

### Methods

The study was performed in 3 steps. Step-1 consisted on HOWL-Q conceptual model construction (literature review and semi-structured interviews to 20 RA outpatients and 20 controls, household workers, who integrated sample (S)-1). Step-2 consisted of instructions selection (by 25 outpatients integrating S-2), items generation and reduction (theory and key informant suggestions, modified natural semantic network technique, and pilot testing in 200 household workers outpatients conforming S-3), items scoring, and questionnaire feasibility (in S-3). Step-3 consisted of construct (exploratory factor analysis) and criterion validity (Spearman correlations), and HOWL-Q reliability (McDonald's Omega and test-retest), in 230 household work outpatients integrating S-4.

### Results

Patients conforming the 4 samples were representative of typical RA outpatients. The initial conceptual model included 8 dimensions and 76 tasks/activities. The final version included 41 items distributed in 5 dimensions, was found feasible and resulted in 62.46% of the variance explained: McDonald's Omega = 0.959, intraclass-correlation-coefficient = 0.921 (95% CI = 0.851–0.957). Moderate-to-high correlations were found between the HOLW-Q,

**Funding:** The authors received no specific funding for this work.

**Competing interests:** The authors have declared that no competing interests exist.

the HAQ, the Quick-DASH and the Lawton-Brody index. HOWL-Q score ranged from 0 to 10, with increasing scores translate into increase limitations.

## Conclusion

The HOWL-Q showed adequate psychometric properties to evaluate household work limitations in women with RA.

## Introduction

Rheumatoid arthritis (RA) is a chronic inflammatory disease, with worldwide distribution and female preponderance, which is extreme in patients from the Latin American region, where the disease is also differentiated by a younger age at presentation [1–4]. RA has a negative impact on patient´s health-related-quality of life (HRQoL) outcomes [5–7], and interferes with the ability to perform job roles and with work productivity. Previous reviews on work disability in developed countries, have shown that 20 to 70% of the RA patients become work disabled within 5–10 years after symptom´s onset [8–11]. The majority of the information related to the impact of rheumatic diseases, including RA, on work disability and productivity has focused on remunerative employment and has been (primarily) assessed with self-report instruments [12–15]. Evidence on psychometric properties and generalizability of the different instruments has been provided to varying degrees. The most recent OMERACT (Outcomes Measures in Rheumatology) recommendations for at work productivity loss measures in rheumatic diseases included 5 instruments and all of them had been applied in patients with RA [14].

According to the International Labour Organization (ILO), domestic work is defined as "work performed in or for a household (ILO Convention 189) [16]. Occupations and tasks considered to be domestic work vary across countries, although, broadly speaking, domestic workers provide personal and household care. The ILO estimates there are 67 million domestic workers, 83% of which are women, ceiling to 92% in Latin America and Caribbean region. Domestic work has unique characteristics, and some of them are of ethical relevance [16]. One conceptual aspect of the ILO definition is that the term domestic workers does not rely on a listing of specific tasks or services that may even change over time, but rather on the common feature that domestic workers work for private households. Based on this conceptual approach, (unpaid) household work is defined by the work (primarily) women do for their families, and the occupations and tasks considered are basically the same as those included under the ILO definition of domestic work.

RA is the most frequent chronic inflammatory arthritis with a clear female preponderance. Household work has been found to be a relevant risk factor for women´s health, particularly when combined with paid work [17, 18] and the burden is heavier in single mothers [18, 19]. In México, up to 33% of female adult aged 15 to 54 years are single-mothers [20]. The WHO report on social determinant of health [21] recommends that (unpaid) household work becomes included in national accounts, due to its large impact on everyday life of millions of people.

Currently, there is not a validated instrument that comprehensively assess the impact of RA on limitations to perform (paid or unpaid) household work, although some of the validated questionnaires assess a limited number of tasks considered to pertain to household work [13, 22]. We decided to consider both household work dimensions, as they share similar tasks and

activities (along the paper the term paid household work will be used as equivalent to the ILO domestic work definition). We consider a tool that evaluates (limitations on) both dimensions is necessary to acquire a comprehensive knowledge about the impact of the disease in the currently predominant female patient´s activities and on their involvement in everyday life situations. Accordingly, the aim of the study was to develop and validate a questionnaire to assess the impact of RA on household work limitations, the HOWL-Q.

## Methods

### Ethical considerations

The study was performed in compliance with the Helsinki Declaration [23]. The Research Ethics Committee of the Instituto Nacional de Ciencias Médicas y Nutrición Salvador-Zubirán (INCMNSZ) approved the study (Reference number: IRE-2702-18-19/1). All the participants agree to participate and gave either written informed consent. Patients and controls that integrated sample 1 (see below) were surveyed to define and rate household tasks/activities; in them, verbal informed consent was approved by local IRB.

### Study design

A 3-step design was used. Step-1 consisted of the HOWL-Q conceptual model construction. Step-2 consisted on instructions and items generation, instructions selection, pilot testing and items reduction, scaling responses and items scoring, and questionnaire feasibility. Step-3 consisted on the HOWL-Q reliability and validity. We followed recommendations for health-related measurement scale´s construction, when a current measure does not exist [24].

### Inclusion/exclusion criteria

Inclusion criteria had mild differences based on the study step. All the patients included were female outpatients with RA, which diagnosis was based on their primary rheumatologist criteria, and no additional rheumatic diagnosis but secondary Sjögren syndrome. Patients that participated in steps 1 and 2 were required to read and write, meanwhile it was not required for patients that participate in Step 3. Also, all the patients included were required to be self-referred household worker but those that participate in instruction´s selection.

 Exclusion criteria considered were additional (to RA diagnosis) uncontrolled comorbid conditions, patients on palliative care and patients who denied to participate.

 Finally, healthy controls were self-referred and absence of any treatment indication was confirmed.

### Samples description

Four different convenience samples (S) of consecutive RA female outpatients were included. In addition, 20 female healthy controls were included in the first sample (S-1). All the patients were recruited from the outpatient clinic of the Department of Immunology and Rheumatology of the INCMNSZ and the 20 healthy controls were identified from the INCMNSZ waiting room and from the blood donor´s bank.

 The S-1 included 40 women (20 RA outpatients and the 20 healthy controls), all of them self-referred as (paid or unpaid) household workers and were selected for the construction of the conceptual model of household work and for item generation. The second sample (S-2) included 25 RA female outpatients, and they participated in the HOWL-Q instruction´s revision process and instruction´s selection. The third sample (S-3) consisted of 200 RA female outpatients, self-referred as household workers who participated in items 'reduction and

HOWL-Q feasibility. Finally, the fourth sample (S-4) consisted of 230 additional RA female outpatients household workers in whom HOWL-Q validity and reliability were assessed.

## Procedures

Procedures that involved patient´s participation were performed the same day patients had a schedule visit to the outpatient clinic of the Department of Immunology and Rheumatology of the INCMNSZ, unless patients required a different timing. At the end of their schedule visit, patients were invited by personnel involved in the study, to a designated area suitable for research purposes. A similar process was followed for healthy control participants.

**Step 1- Construction of the conceptual model of the HOWL-Q.** *Literature review*. Three authors (VPR, AOH and ICY) reviewed the literature and identified: (A) Published relevant and validated tools to assess work disability and (limitations in) work productivity in patients with RA and/or additional rheumatic diseases; (B) activities considered by the Mexican National Institute of Statistics and Geography (Instituto Nacional de Estadística, Geografía e Informática, INEGI) as household work; and, (C) the potential evaluative dimensions of the household work.

The majority of the tasks/activities considered by INEGI as household work were under the category of instrumental activities of daily living [25]; accordingly, we also identified relevant and validated instruments/scales that assess (limitations) in such activities, with a particular focus on chronic/rheumatic diseases.

*Patient proposals*. Previously, the different household work dimensions were identified; then, patients and healthy controls from S-1 were asked to list and rate (according to difficulty) the most difficult household tasks/activities to perform in the face of physical disability. For each dimension, a stimulus sentence was given to participants and each task/activity provided by them, would correspond to a defining word.

**Step 2-Instructions and items generation, instructions selection, pilot testing and items reduction, scaling responses and items scoring, and questionnaire feasibility.** *HOWL-Q instructions and items development, and their selection and reduction*. Two types of sources were considered for item´s generation: Theory and key informant suggestions. In addition, two authors, blinded to each other proposals, drafted (two each) four different versions of the HOWL-Q instructions; instructions were intended to assess the degree of difficulty in carrying out the tasks/activities included in each dimension. Patients from S-2 evaluated these versions, and were directed to select only one.

For items generation and reduction, the totality of the defining words provided by the participants from S-1, were analyzed using the modified natural semantic network technique [26, 27]. The words that were above the breakpoint of Catell were introduced as activities, one activity per item, into the first draft/version of the instrument (v1).

Then, v1 was subjected to a first evaluation by and experts-committee, conformed by two rheumatologists, one occupational doctor, two psychiatrists, two nurses assigned to an early rheumatoid arthritis clinic, two social workers from de Department of Immunology and Rheumatology, a sociologist and finally, an anthropologist. Experts were directed to evaluate instructions and items, according to the following criteria: Adequate wording, appropriate language and meaning for the target population and absence of affective load (limited to items). Instructions and items with negative evaluations were modified. After this revision, a second draft/version was obtained (v2), which was applied to patients from S-3, during a pilot test. Items that fulfilled the consecutive criteria/steps were eliminated (Please refer to the S1 Appendix "S1 Table. Consecutive steps/criteria for item´s reduction", S1 Appendix) and HOWL-Q v3 was derived which was applied during validation process.

*Scaling responses.* We selected a direct estimation method of responses, on a 4-point Likert-scale, where 0 = without difficulty, 1 = with some difficulty, 2 = with much difficulty and 3 = unable to do the task/activity. In addition, we added the "not applicable" option.

*Items scoring and interpretation.* HOWL-Q scoring proposed was similar to that recommended for the Health Assessment Questionnaire Disability Index, (HAQ-DI) [28].

*Feasibility.* Feasibility was evaluated in S-3, according to the following criteria: Time required to fill the HOWL-Q, patient's perceived instructions and items clarity, and patient´s format acceptance.

**Step 3. Psychometric validation of the v3 HOWL-Q.** Construct validity was evaluated for each questionnaire dimension using factor analysis. Criterion (convergent) validity was based on correlations between the HOLW-Q score and the HAQ-DI (Health Assessment Questionnaire- Disability Index) score [28], the Lawton-Brody index score [25] and the Quick DASH (Quick Disability of the Arm, Shoulder and Hand) score [29], all of them applied concomitant to the HOWL-Q. In addition, in RA females with paid household work, correlation between the HOWL-Q and the WPAI (Work Productivity and Activity Impairment) questionnaire [30] was examined (see description below).

Reliability was examined with internal consistency and temporal stability; test-retest was performed in 50 patients from S-4, in whom HOWL-Q was applied twice, within a 2±1 week's interval.

## Instruments description

A brief description of the HAQ-DI, the Lawton-Brody Index, the Quick DASH and the WAPI is summarized in **Table 1**. All of them are self-assessment questionnaires/index, developed in adults.

## Statistical analysis

For step 1, the modified natural semantic network technique was used and published recommendations followed as above referred [26, 27].

For step 2, descriptive statistic was generated for each item. The positive bias was evaluated by kurtosis value and the discriminant capacity of the items was evaluated by t test. The homogeneity of the items was tested with the inter items correlation and the items' contribution to the total score. Items with kurtosis value between +0.5 and -0.5, with t test (and F test) p value > 0.05 and with correlations > 0.8 were discarded [24, 31]. Face and content validity by experts was examined with agreement percentage.

For step 3, McDonald's Omega and inter-item correlation for the complete scale and for each dimension was used to assess the internal consistency of the questionnaire. McDonald's

**Table 1. Description of instruments.**

| Instruments | Description | N° of items | Scoring and interpretation |
|---|---|---|---|
| HAQ-DI | Assesses functional status. | 20 | Summary score from 0 to 3, with increasing scores indicating higher disability. |
| Lawton-Brody Index | Assesses a person's ability to perform instrumental activities of daily living. | 8 | Summary score from 0 (low function, dependent) to 8 (high function, independent). |
| Quick DASH | Assesses symptoms and function of the entire upper extremity. | 30 | Cut-off scores: <15 = "no problem," 16–40 = "problem, but working," and >40 = "unable to work". |
| WPAI | Examines the extent of absenteeism, presenteeism, and impairment in daily activities attributable to general health or a specific health problem. | 6 | Summary score expressed as % of impairment/productivity loss, with higher scores indicating greater impairment. |

Omega interpretation was as follows: <0.70 indicates that individual items provide an inadequate contribution to the overall scale and values of >0.90 suggest redundancy [32]. For test–retest, intra-class correlation coefficients (ICC) and their 95% confident intervals (CI) were calculated based on a single measurement, absolute-agreement, 2-way mixed-effects model. According to the ICC, values <0.5 indicate poor reliability, between 0.5–0.75 moderate reliability, between 0.75–0.9 good reliability and values >0.9 indicate excellent reliability. Finally, 95% CI estimates between 0.83–0.94 were considered as good reliability level and those between 0.95–0.99 estimates, as excellent reliability level [33]. Floor and ceiling effects were determined as the percentage of patients who achieved the lowest and highest score of the scale, respectively. Construct validity was evaluated using exploratory factor analysis (principal components) with direct oblimin rotation. Sampling adequacy was confirmed using the Kaiser-Mayer-Olkin (KMO) (appropriate value ≥0.5) measure, and the use of factor analysis was supported by Bartlett's test of sphericity (significant value p<0.05). The number of factors was determined as the number of eigenvalues >1. The item-factor membership was determined by the factor loading as an indication of the degree to which each item was associated with each factor [34].

Criterion (convergent) validity was analyzed using Spearman rank correlation coefficient (rho) to determine the strength of the relationship between the HOWL-Q global score and specific dimension scores, and the HAQ-DI (global score and specific dimensions scores), the Lawton-Brody index (global score) and the Quick-DASH questionnaire (global score); in the subpopulation of women from S-4 with paid household work, the relationship between the HOWL-Q global score and specific dimension scores with specific dimensions from the WAPI questionnaire was also examined. The strength of the correlation was interpreted as high (rho > 0.7), moderate (0.4 to 0.7) and low (<0.4), [24].

Sample´s sizes were based on the methodological recommendations, which suggested a minimum of 50 patients for assessing construct validity, a minimum of 100 patients for assessing internal consistency, and 5 to 10 patients for each item of the instrument [35].

Descriptive statistics was performed to describe the socio-demographic and clinical characteristics of the patients and healthy controls included in the 4 samples.

All statistical analyses were performed using Statistical Package for the Social Sciences version 21.0 (SPSS Chicago IL). A value of p<0.05 was considered statistically significant.

## Results

### Sample´s description

The 475 female RA outpatients included, were divided in four samples, and were representative of typical RA patients attending the outpatient clinic of our Institution, middle-aged women, with long-standing disease and disease-specific autoantibodies. Their data are summarized in **Table 2**. In general, patients and diseases characteristics were similar across samples, albeit some differences were observed: patients from S-4 referred higher years of formal education compared to patients from S-1, S-2 and S-3 (p≤0.001), and had lower serum values of CRP when compared to patients from S-2 and S-3 (p≤0.001); patients from S-4 were more frequently self-referred unpaid household workers compared to patients from S-1, S-2 and S-3 (p≤0.001); patients from S-2 were more frequently in clinical remission compared to patients from S-3 and S-4 (p≤0.002); finally, patients from S-3 received more frequently an indication for joint replacement/had joint replacement when compared to patients from S-2 and S-4 (p≤0.02).

The 20 healthy controls who integrated S-1 had median (IQR) age of 42 years (35.3–56), formal education of 10.5 years (9–13), were primarily married or living together (60%); meanwhile, 50% were self-referred paid household workers.

**Table 2. Description of the samples´ characteristics.**

| | S-1[1], N = 20 | S-2, N = 25 | S-3, N = 200 | S-4, N = 230 |
|---|---|---|---|---|
| Years of age | 54 (51.3–60.5) | 56.9 (43.3–61.2) | 53.3 (44–62) | 55 (44.8–63) |
| Years of formal education | 9 (6–12) | 9 (6–9) | 9 (7.3–12) | 11 (6–16) |
| Patients with medium-low SE level[2] | NA | 31 (92) | 185 (92.5) | 203 (88.3) |
| Patients married or living together[2] | 12 (60) | 17 (68) | 103 (51.5) | 126 (54.8) |
| Patients self-referred unpaid household workers[2] | 13 (65) | 17 (68) | 111 (55.5) | 223 (97) |
| Years of disease duration | NA | 14 (10–17) | 17 (10–23.6) | 15 (8–24) |
| Patients with early disease[2] | NA | 0 | 19 (9.5) | 29 (12.6) |
| Patients with clinical remission[2] | NA | 13 (52) | 42 (21) | 49 (21.3) |
| CRP, mg/dL | NA | 0.84 (0.26–4.2) | 0.6 (0.23–1.61) | 0.45 (0.26–1.19) |
| Patients with RF[2] | NA | 24 (96) | 196 (98.5) | 226 (98.3) |
| Patients with ACCP[2] | NA | 24 (96) | 195 (98) | 228 (99.1) |
| Patients with major comorbidities[2] | NA | 4 (16) | 34 (17) | 31 (13.5) |
| Patients with indication for joint replacement/joint replacement[2] | NA | 5 (20) | 94 (47) | 43 (18.7) |

Data presented as median, IQR unless otherwise indicated

[1] Only characteristics from the RA patients are presented in the table.

[2] Number (%) of patients. NA = Data not available.SE = socio-economic. CRP = C reactive protein. RF = rheumatoid factor. ACCP = antibodies to cyclic citrullinated peptides.NA = not available.

In S-4, the majority of the patients were self-referred as unpaid household workers and spent (median, IQR) 28 (21–49) hours/week taking care of their family and home; meanwhile, the seven patients left declared to work as (paid) household worker and to additionally spent 21 (14–35) hours/week taking care of their family and home.

## Step 1- Construction of the conceptual model of the HOWL-Q

**Literature review.** We identified several tools to assess work disability and (limitations in) work productivity in patients with RA and/or additional rheumatic diseases; none of the instruments/scales was specific and validated to assess the impact of RA on (paid) household work. Also, we identified a questionnaire that assess limitations in (basic and) instrumental activities of daily living in RA patients [36] and one additional scale that focus on instrumental activities of (geriatric) women with chronic illness [25].

We adopted the INEGI conceptual model of household work that considers 6 dimensions (1-food preparation and food serving, 2-cleaning and care of clothes and footwear, 3-shopping and home administration, 4-cleaning and maintenance of housing, 5-care and support for household members, and 6-help other homes and volunteer [20], but split the cleaning and maintenance of housing dimension into inside and outside cleaning; in addition, we adopted and added the dimension of "Transportation" which is included in the tool that assesses functional limitations in instrumental activities of daily living in women with chronic diseases [25]. The final theoretical conceptual model included the 8 dimensions (**Fig 1**).

## Step 2. Instructions and items generation, instructions selection, pilot testing and items reduction, scaling responses and items scoring

**Patients and healthy controls proposals (S-1) for items generation.** Semantic network technique identified 162 defining words, corresponding to tasks/activities of household work; based on the frequency analysis, 76 activities were included in the 8 dimensions of the final household construct and v1 of the HOWL-Q was integrated with 76 activities.

**Fig 1. Theoretical household work construct.** Tasks/activities deleted from version 1 and 2.

*Experts-committee evaluation/selection of the instructions and of the items*. Results of 11 expert's evaluations, for each of the 76 items/activities (N = 836 evaluations) are summarized in **Table 3**: Twenty items had some kind of observations by at least one member from the experts-committee; in all the cases, expert´s suggestions were incorporated and items rephrased. Finally, 10 experts (91%) agree on instructions´ clarity. The HOWL-Q v2 was generated.

**Pilot testing and Items' reduction.** The v2-HOWL-Q was applied to the 200 outpatients integrating S-3. Twenty questionnaires were incomplete (10%) and returned back to the patients who were asked to fill missing data. There was no consistency in the items with omitted response.

Items evaluation showed six items without discriminant capacity, seven items with a frequency < 5% in any of the response options, six items with correlations > 0.8, nine items with factor loading < 0.5 and two items with factor loading > 1 factor. The total number of items discarded was 30 and the v3 of the HOWL-Q was derived, which included 46 tasks/

Table 3. Expert's evaluations.

| Criteria evaluated | Number (%) of expert-evaluations with the criteria |
| --- | --- |
| **Items** | |
| Adequate writing | 743 (89) |
| Appropriate language/meaning for the target population | 761 (91) |
| Absence of affective load | 806 (96.4) |

Table 4. HOWL-Q versions 'structure previous validation process.

| Dimensions | N° of tasks/activities | | |
|---|---|---|---|
| | Version 1 | Version 2[a] | Version 3[b] |
| 1.- Food preparation | 6 | 6 | 6 |
| 2.- Care of clothes | 11 | 11 | 9 |
| 3.- Shopping and home administration | 8 | 8 | 6 |
| 4.- Housekeeping inside the house | 15 | 15 | 10 |
| 5.- Housekeeping outside the house | 11 | 11 | 2 |
| 6.- Caring for children | 9 | 9 | 6 |
| 7.- Caring for people with disabilities | 7 | 7 | |
| 8.- Transportation | 9 | 9 | 7 |

[a]Version 2 included same number of dimensions and of items than version 1, but some items were rephrased.

[b]Dimensions 6 and 7 were joined and renamed "Caring for people" (children, seniors and/or people with disabilities).

activities of household work distributed in seven dimensions (dimensions 6 and 7 were joined), as summarized in **Table 4**.

**Item scoring.** We proposed a HOWL-Q scoring system based on 3 steps: At first, to calculate the score of each dimension, assigning the dimension the highest individual item score; at second, sum the dimension scores and divide the sum by the number of dimensions answered; finally, transform the final score on a 10-point scale. The possible global HOWL-Q score ranged from 0 to 10. Increasing scores translate into increase household work limitations.

In addition, we agree that in order to provide a global HOWL-Q score, the five dimensions will be required but for patients who do not care for others; in those patients the HOWL-Q score is calculated considering the 4 dimensions left. Dimensions could be scored if at least ≥1 item had one option selected. Accordingly, (median, IQR) global HOWL-Q score for S-4 was 5.3 (3.3–7.3). S2 Appendix "S2 Table. Descriptive statistics for individual items of the HOWL-Q" (S2 Appendix) summarizes HOWL-Q individual items descriptive statistics.

**Evaluation of the HOWL-Q feasibility.** HOWL-Q feasibility was assessed in S-3. Mean time to fill the questionnaire was of 30 minutes; the majority of the patients (60%) rated the time to fill the v2 HOWL-Q as considerable. Patients agreed about instructions clarity (100%) and about item´s clarity (100%), and 95% of them agreed about the adequacy of the v2 HOWL-Q format.

## Step 3. Psychometric validation of the HOWL-Q

**Validity.** *Construct validity. Construct validity* was evaluated with factor analysis and results are summarized in **Table 5**. The KMO measure was of 0.957 and significant result ($X^2$ = 8054.293, p≤0.001) for the Bartlett sphericity test confirmed the adequacy of the sample. A 5-factors structure was extracted, which accounted for 62.46% of the total variance. All factors had eigenvalues >1. The post validation structure of the questionnaire differed from the initial proposal and is represented in **Fig 2**. The final version of HOWL-Q included 41 items in 5 dimensions equivalent to the 5 factors extracted (Please refer to the S3 Appendix).

*Criterion validity (convergent validity).* **Table 6** summarizes Spearman rank correlation coefficient (rho) between the HOWL-Q global score and specific dimensions scores, and the HAQ-DI (global score and specific dimensions scores), the Lawton-Brody Index (global score), the Quick-DASH questionnaire (global score) and the four dimensions of the WPAI questionnaire (in the subpopulation of RA females with paid household work); the strength of the correlations was moderate to high, and consistently significant between the HOWL-Q and

**Table 5. Factor loadings for the 5 factors after direct oblimin rotation of the HOWL-Q.**

| D1 | HOUSEKEEPING | 1 | 2 | 3 | 4 | 5 |
|---|---|---|---|---|---|---|
| | 5.1_Outdoor sweeping. | .661 | | | | |
| | 4.1_Mopping. | .651 | | | | |
| | 2.5_Hanging clothes. | .645 | | | | |
| | 2.8_Folding clothes. | .623 | | | | |
| | 4.2_Changing sheets or making the bed. | .616 | | | | |
| | 5.8_Dusting. | .591 | | | | |
| | 4.12_Washing dishes. | .573 | | | | |
| | 2.11_Carry wet clothes. | .545 | | | | |
| | 4.15_Changing curtains. | .510 | | | | |
| | 2.7_Ironing clothes. | .487 | | | | |
| | 3.8_Write. | .487 | | | | |
| | 4.9_Cleaning or washing the toilet or bathroom walls. | .407 | | | | |
| D2 | INTERACTION WITH OBJECTS AND PERSONS | | | | | |
| | 2.2_ Carrying heavy objects. | | .837 | | | |
| | 4.3_ Moving heavy objects or furniture. | | .777 | | | |
| | 1.2_ Lifting or carrying heavy objects. | | .718 | | | |
| | 3.4_ Pushing the shopping cart. | | .708 | | | |
| | 3.1_ Carrying bags or heavy objects. | | .707 | | | |
| | 2.10_ Carrying water buckets. | | .552 | | | |
| | 6.1_ Carrying or mobilize, some (child, elderly or sick) who is at your charge or care. | | .512 | | | |
| | 2.3_ Washing clothes by hand. | | .411 | | | |
| D3 | CARE FOR OTHERS | | | | | |
| | 6.6_ Feeding someone (child, elderly or sick) who is in charge or care. | | | .730 | | |
| | 6.9_ Supporting to carry out tasks, to some person (child, elderly or sick) who is in charge or care. | | | .711 | | |
| | 6.3_ Supporting someone to dress (child, elderly or sick) in charge or care. | | | .622 | | |
| | 6.7_ Combing the hair of any person (child, elderly or sick person) who is in charge or care. | | | .541 | | |
| | 6.8_ Supporting or accompany in recreational activities or not, any person (child, elderly or sick) who is in charge or care. | | | .535 | | |
| | 3.2_ Buying food or grocery shopping. | | | .496 | | |
| D4 | PERSONAL MOTION AND TRANSPORTATION | | | | | |
| | 8.1_ Getting on or off (bus, subway, bike, taxi). | | | | .831 | |
| | 1.5_ Standing. | | | | .729 | |
| | 4.14_ Climbing stairs or onto benches. | | | | .695 | |
| | 8.4_ Sitting down or getting up. | | | | .625 | |
| | 3.3_ Walking somewhere to do some diligence. | | | | .604 | |
| | 4.8_ Collecting objects from the floor. | | | | .575 | |
| | 8.7_ Standing during public transport. | | | | .544 | |
| | 8.9_ Walking or moving on the bus. | | | | .492 | |
| D5 | ACTIVITIES REQUIRING MANUAL DEXTERITY | | | | | |
| | 3.5_ Receiving change (coins). | | | | | .639 |
| | 1.6_ Open the oven or turn on the stove knobs or similar. | | | | | .544 |
| | 1.1_ Peeling, cutting or chopping food. | | | | | .543 |
| | 2.6_ Patching or repairing clothes. | | | | | .539 |
| | 8.6_ Opening car doors. | | | | | .526 |
| | 1.3_ Opening or closing cans and jar lids. | | | | | .525 |
| | 2.1_ Wringing out clothes. | | | | | .514 |

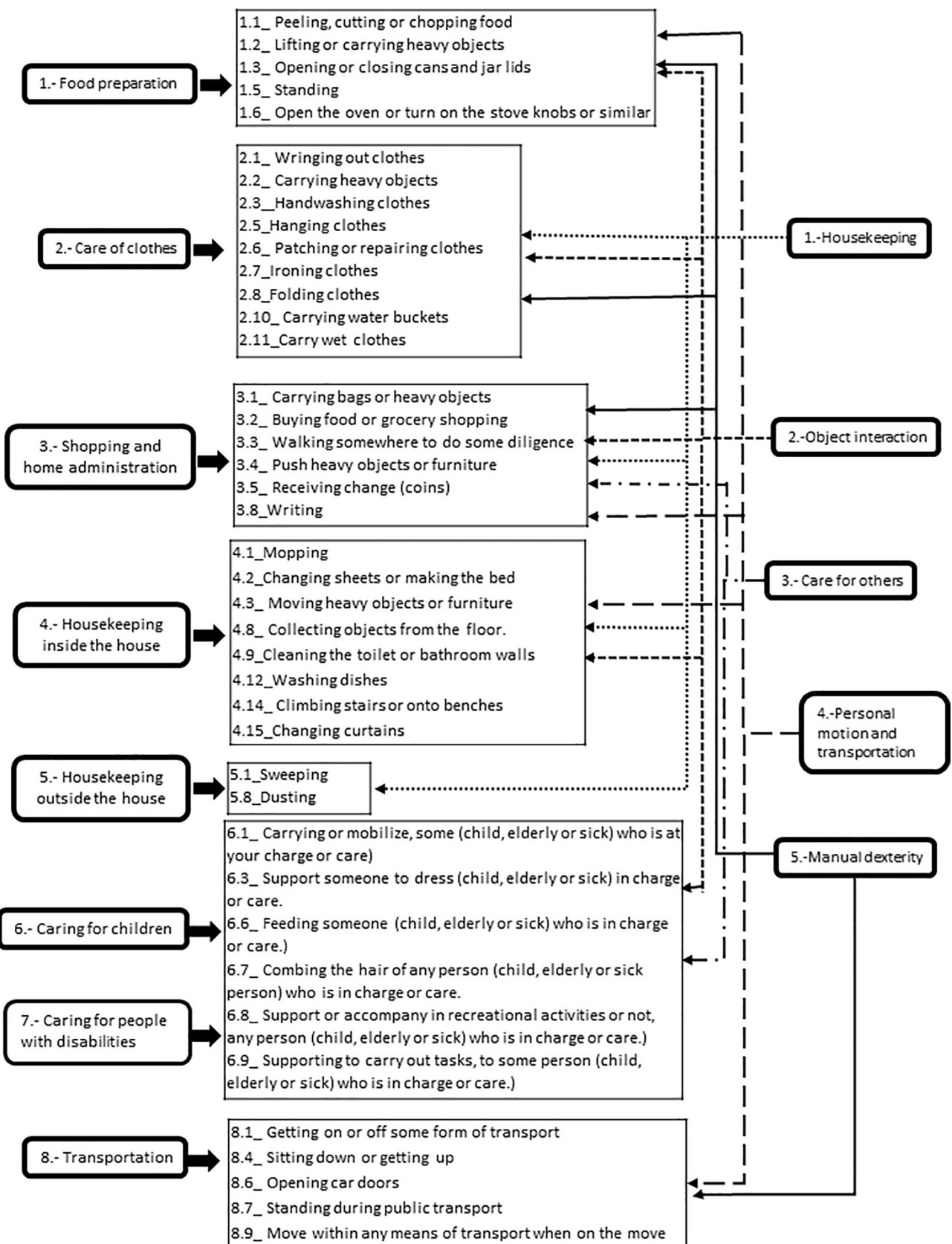

**Fig 2. Pre and post-validation HOWL-Q structure.**

**Table 6. Strength of the association between the HOWL-Q global score and individual dimensions scores, and the HAQ-DI, the Lawton-Brody Index, the Quick-DASH questionnaire and the WPAI questionnaire.**

|  | HOWL-Q | HOWL-Q D1 | HOWL-Q D2 | HOWL-Q D3 | HOWL-Q D4 | HOWL-Q D5 |
|---|---|---|---|---|---|---|
| **HAQ-DI** | 0.797** | 0.687** | 0.595** | 0.629** | 0.739** | 0.671** |
| Dresssing and groomig | 0.612** | 0.547** | 0.439** | 0.552** | 0.534** | 0.468** |
| Arising | 0.661** | 0.569** | 0.476** | 0.514** | 0.655** | 0.525** |
| Eating | 0.639** | 0.559** | 0.517** | 0.439** | 0.526** | 0.629** |
| Walking | 0.591** | 0.492** | 0.400** | 0.546** | 0.591** | 0.431** |
| Hygiene | 0.639** | 0.540** | 0.482** | 0.518** | 0.631** | 0.488** |
| Reach | 0.623** | 0.544** | 0.487** | 0.447** | 0.577** | 0.542** |
| Grip | 0.540** | 0.437** | 0.420** | 0.406** | 0.438** | 0.574** |
| Common daily activities | 0.706** | 0.623** | 0.496** | 0.580** | 0.677** | 0.579** |
| **Quick Dash questionnaire** | 0.746** | 0.643** | 0.572** | 0.619** | 0.647** | 0.640** |
| **Lawton-Brody index** | -0.470** | -0.412** | -0.371** | -0.328** | -0.402** | -0.444** |
| **WPAI‡ questionnaire** |  |  |  |  |  |  |
| Absenteeism | 0.774* | 0.838* | 0.424 | 0.893** | 0.419 | 0.320 |
| Presenteesism | 0.746 | 0.727 | 0.539 | 0.182 | 0.894* | 0.874* |
| Work productivity loss | 0.987** | 0.985* | 0.531 | 0.391 | 0.851* | 0.845* |
| Activity Impairment | 0.810* | 0.769* | 0.428 | 0.118 | 0.962* | 0.970* |

**p = 0.001

*p = 0.005

‡Correlations established only in 7 patients who performed paid household work. HOWL-Q = household work limitation questionnaire; D = dimension (of the HOWL-Q); HAQ-DI = Health Assessment Questionnaire- Disability Index; WPAI = Work Productivity and Activity Impairment.

the HAQ-DI, the Quick-Dash questionnaire and the Lawton-Brody index. Of note, there were only 7 women who referred paid household work, in whom the strength of the association between the HOWL-Q and the WPAI questionnaire was found significant and high with absenteeism and work productivity loss dimensions.

**Reliability.** Results of *internal consistency* (McDonald´s Omega) and temporal stability/test-retest (ICC and 95% CI) for each dimension of the final version of the HOWL-Q are presented in **Table 7**, which additionally includes inter-item correlation and floor and ceiling effects.

The mean (±SD) of the time between the 2 measurements in the test-retest analysis was of 25.04 (±4.34) days. *Temporal stability* is summarized in Table 7.

## Discussion

Household work remain predominately the domain of women; while the paid dimension of household work may provide a valuable entry point into the labour market for women, the

**Table 7. Psychometric characteristics dimensions that integrated the final version of HOWL-Q.**

| HOWL-Q Dimensions (N° if items) | McDonald's omega | ICC 95% CI | Mean of inter-item correlations | Floor / ceiling effects (%) |
|---|---|---|---|---|
| Dimension one (12) | 0.851 | 0.952 (0.914–0.973) | 0.530 | 12.2 / 24.3 |
| Dimension two (8) | 0.860 | 0. 910 (0.842–0.949) | 0.592 | 4.3 / 41.7 |
| Dimension three (6) | 0.779 | 0.824 (0.685–0.901) | 0.593 | 23.5 / 14.3 |
| Dimension four (8) | 0.848 | 0.880 (0.775–0.934) | 0.536 | 12.6 / 15.2 |
| Dimension five (7) | 0.750 | 0.899 (0.821–0.943) | 0.471 | 6.1 / 17.4 |
| HOWL-Q (41) | 0.959 | 0.921 (0.851–0.957) | NA | 2.2 / 0 |

HOWL-Q = household work limitation questionnaire. ICC 95% CI = intra-class correlation coefficients and their 95% confident intervals. NA = not applicable.

downside is that poor working conditions and insufficient legal protection disproportionately affect women and reinforce gender disparities in relation to access to "decent" work [12]. In the Latin-America and the Caribbean region, the paid dimension of household work has differential characteristics such as a rapid growth of the sector, a particular pattern of migration and an over representation of international migrants and of women with lower formal education attainment [16, 37]. The majority of these characteristics are well recognized sources of both, extrinsic and intrinsic vulnerability [38]. Meanwhile, the unpaid housework dimension, remains an invisible contribution of women to the economic survival of the families and to the development of homes.

Since 2013, there is an increased need to recognize the impact of the different works into individual, family and societal developments, and the unpaid dimension of the household work has been proposed to be measured in order to make it visible [1, 16, 39, 40]. This challenge emerges as particularly relevant in the clinical context of RA, a chronic disease that primarily affects middle aged women, where the literature has consistently evidenced a picture of impaired functioning of basic and independent activities of daily living and of work productivity [6, 8–11, 41, 42]. Existing literature has also reported significant limitations in RA patient´s ability to carry out tasks and activities that define household work although with a limited number of studies and tasks/activities evaluated [43, 44]. Backmanm et al [43] developed a mail survey in consultation with 18 working-age adults with RA, in order to define determinants of participation in paid and unpaid work by adults with RA; the survey was mailed to 269 community-dwelling adults with RA; authors reported a recruitment response rate of 40% and found that work limitation affected both paid and unpaid work; interestingly, factors associated with greater participation in paid work, differed from those associated with greater participation in unpaid work. Reisine et al [45] were the firsts to quantify the degree of limitation in instrumental and nurturant role dimensions, and to describe the associate satisfaction with functioning in these role dimensions, in 142 women with RA, living with a husband, dependent children(s), or both, at the time of disease onset; authors found that both area were affected by RA, with only 5% of the patients included indicating no limitation in the functions studied. Allaire et al [44] developed the household work performance questionnaire based on the model proposed by Reisine et al [45]; the questionnaire was applied to patients with RA and authors demonstrated that household work disability was present in women with RA and that health related factors were the strongest predictor of disability, although family and personal factors also had significant effects.

Due to the absence of validated instruments to comprehensively evaluate the complex construct of household work, we developed the HOWL-Q, which was designed to assess RA-related limitations to perform (paid or unpaid) household work. According to the ICF (the International Classification of Functioning, Disability and Health) and in the context of health, activity limitations are defined as difficulties an individual (with a health condition) may have in executing activities [46]. The household work construct was integrated with both instrumental and nurturing activities. Sidney Katz describe the concept of activities of daily living to refer to people´s daily self-care activities [47]; since then, the activities under this umbrella term has grown and been redefined, although health professionals often use a person ability or inability to perform activities of daily living as a measurement of their functional status. Meanwhile, instrumental activities of daily living are not necessary for fundamental functioning, but they let an individual live independently in a community, and are considered on a somewhat more complex level of organized human behavior than the former group of activities [47]; shopping, cooking, housekeeping, laundry, use of transportation, managing money and medication, and the use of telephone are among the classic instrumental activities of daily living.

Currently, there is documented evidence that the evaluation of these functions helps to identify problems that require treatment or care and produces information about prognosis.

HOWL-Q development was rigorous from a methodological perspective. The construct of household work was supported by a systematic literature review and reference data from a national Institution, the INEGI, were also considered. In addition to healthcare professionals, household workers patients with RA and healthy controls contribute to reframe the construct under development. Whereas clinicians may be the best observers for certain RA aspects such as manifestations, only patients with the disease can report on more subjective elements [48]. Representative populations of RA patients were accordingly included for the HOWL-Q development, in order to enhance content validity and to ensure that the patients experience is captured. Semi-structured interviews were conducted by 2 researchers trained in interview procedures, meanwhile qualitative analysis was performed by a PhD psychologist; accordingly, step-1 successfully identified household work-related tasks and activities that patients considered their priorities. Cognitive interviews are considered crucial during patient-reported outcomes measures development, as highlighted in published guidelines [49, 50]. In addition, the wording for single items to capture each of these, was optimized through a process of face validity testing with a multidisciplinary group of healthcare providers with experience in the management of RA patients. Meanwhile, instruction´s wording and clarity was submitted to patient´s evaluations, as was the final version of the HOWL-Q, prior to its formal validation process; the changes subsequently made were crucial for enhancing acceptance by the intended population and to prevent future inaccurate data collection. Finally, the selection and reduction of the items was performed by evaluating the ability to discriminate the extreme values of each patient´s selection, the distribution of responses in order to verify that all modalities of responses were used, and the presence of floor and ceiling effect. Importantly, 4 different samples of RA patients were used; patients included were representative of typical RA outpatients.

After the validation process, the structure of the HOWL-Q underwent a modification from the initial conceptual model; the 41 items retained were distributed in 5 dimensions, which were named as follows: housekeeping, interaction with objects and persons, care for others, personal motion and transportation, and activities requiring manual dexterity. The initial structured was based on household work tasks and activities, while the final structure derived from incorporating the notion of ability/inability to perform the tasks and activities pertaining to household work.

Regarding its psychometric properties the HOWL-Q showed adequate internal consistency; McDonald´s Omega coefficient for the total scale and individual dimensions were excellent [32]. The test-retest reliability assessed in 50 patients by the same researcher showed an ICC and 95% CI indicating good to excellent reliability [36]. The construct validity was demonstrated by KMO sampling and Barlett´s test of sphericity, both confirming the adequacy of the sample size for conducting factor analysis [34], and a single factor structure was extracted, accounting for 62.46% of the variance. Face and content validity of the HOLW-Q version 2 were examined by a multidisciplinary group of experts involved in RA management. The moderate-to-high Spearman correlations between HOWL-Q scores and additional questionnaires that assess limitations in basic and instrumental activities favored criterion validity. Patients confirmed HOWL-Q feasibility of the version 3. Finally, the global score of HOWL-Q did not show neither floor nor ceiling effect, which had been defined when more than 15% of the patients achieved the lowest or highest score, respectively [35]; both, floor and ceiling effects could reduce the possibility of detecting change over time.

Finally, we proposed a HOWL-Q total score to ensure simplicity in routine clinical practice, which emerged as important; the highest individual item score was assigned to each dimension and scores from dimensions were summed together and the sum divided by 5 (or 4 in case of not realizing care for others). Alternative scoring approaches were considered including

weighting items but it was agreed that a non-weighted approach was better suited to using the tool in everyday practice. In addition, we are proposing a single score which may help to evaluate the individual impact of RA on household work, meanwhile, each dimension is additionally score in order to facilitate a comprehensive analysis. Also, we propose that the 5 dimensions need to be scored with at least one item to guaranty that the household work construct is assessed. The median HOWL-Q score for our population was 5.3 which confirms that our target population had some level of household work limitation, which is expected in a population with long-standing disease and medium-low socioeconomic status.

Limitations of the study need to be addressed. First, the study was performed in Spanish speaking patients from one Mexican-based rheumatology outpatient department, which may affect generalizability of the results; in addition, any translation from the HOWL-Q should be done in concordance with good practices [51]. A second limitation relates to interviews which were performed in a controlled research environment that differs from the real clinical setting. A third limitation is related to the household work construct and particularly to the emotional dimension which was not considered in the current work; although care for children and the elderly were identified and integrated in an individual dimension, there is literature that documents that household workers additionally contribute to the "mental hygiene" of the family [52]. Forth, the HOWL-Q was developed to assess the impact of RA on household work limitations; the concept of impact in patients with chronic conditions (such as RA), was developed by patients and researchers and termed the "impact triad"; participants proposed that it should combine the severity of an outcome, its importance to patients and their ability to self-manage it [53]. Fifth, only seven patients form S-4 were self-referred paid household workers which limits convergent validity assessment between the HOWL-Q and the WPAI. Six, work is ultimately a function of both the person and his/her work context (like the environmental factors) and the manner which they interact [54] and these components along with their interaction may differ in the paid and unpaid household work dimensions and within countries. Finally, we did determine a limited number of psychometric characteristics of the HOWL-Q, those considered necessary for a first approach; additional relevant characteristics such as sensitivity to change and values within normal values need to be defined.

## Conclusions

In this paper we presented the rationale and the methods for the development and validation of a questionnaire allowing for assessing the impact of RA on household work limitations. The questionnaire can be easily implemented in the outpatient setting and for research purposes.

In RA, there has been some research investigating social functioning in terms or work roles with relatively little attention given to other social roles such as domestic work; this oversight is particularly surprising given the higher prevalence of RA among middle-aged women, particularly in the Latin-American region. Importantly, 2 types of work are fundamental to capitalist societies: paid employment associated with the waged economy and unpaid domestic work that produces and sustains both the current generation of workers and the children who are the future work force [55]. For a variety of reasons, women tend to spend more time on unpaid household work than men and this figure is extreme in Latin-America. The HOWL-Q offers the possibility to make visible household work and to adopt a more comprehensive approach of the impact of RA on women´s life.

## Supporting information

**S1 Appendix. Consecutive steps/criteria for item´s reduction.**
(PDF)

**S2 Appendix. Descriptive statistics for individual items of the HOWL-Q.**
(PDF)

**S3 Appendix. Final draft of the HOWL-Q, reduced to 41 items.**
(PDF)

## Author Contributions

**Conceptualization:** Ana Belén Ortiz-Haro, Abel Lerma-Talamantes, Irazú Contreras-Yáñez, Virginia Pascual-Ramos.

**Formal analysis:** Ana Belén Ortiz-Haro, Abel Lerma-Talamantes, Irazú Contreras-Yáñez.

**Investigation:** Ana Belén Ortiz-Haro, Ángel Cabrera-Vanegas, Irazú Contreras-Yáñez.

**Methodology:** Abel Lerma-Talamantes, Irazú Contreras-Yáñez, Virginia Pascual-Ramos.

**Project administration:** Virginia Pascual-Ramos.

**Software:** Abel Lerma-Talamantes.

**Supervision:** Ana Belén Ortiz-Haro, Virginia Pascual-Ramos.

**Writing – original draft:** Irazú Contreras-Yáñez, Virginia Pascual-Ramos.

**Writing – review & editing:** Ana Belén Ortiz-Haro, Abel Lerma-Talamantes, Ángel Cabrera-Vanegas, Irazú Contreras-Yáñez, Virginia Pascual-Ramos.

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
