## [Decision Letter · Decision Letter 0]

4 May 2020

PONE-D-20-04057

Development and validation of a questionnaire assessing household work limitations (HOWL-Q) in women with rheumatoid arthritis.

PLOS ONE

Dear Dr Pascual-Ramos,

Thank you for submitting your manuscript to PLOS ONE. After careful consideration, we feel that it has merit but does not fully meet PLOS ONE’s publication criteria as it currently stands. Therefore, we invite you to submit a revised version of the manuscript that addresses the points raised during the review process.

We would appreciate receiving your revised manuscript by Jun 18 2020 11:59PM. To enhance the reproducibility of your results, we recommend that if applicable you deposit your laboratory protocols in protocols.io, where a protocol can be assigned its own identifier (DOI) such that it can be cited independently in the future. For instructions see: http://journals.plos.org/plosone/s/submission-guidelines#loc-laboratory-protocols

We look forward to receiving your revised manuscript.

Kind regards,

Luca Navarini

Academic Editor

PLOS ONE

Journal Requirements:

2. Please amend your current ethics statement to address the following concerns: Please explain why written consent was not obtained, how you recorded/documented participant consent, and if the ethics committees/IRBs approved this consent procedure.

3. We note you have included a table to which you do not refer in the text of your manuscript. Please ensure that you refer to Table 6 in your text; if accepted, production will need this reference to link the reader to the Table.

Reviewers' comments:

Reviewer's Responses to Questions

**Comments to the Author**

1. Is the manuscript technically sound, and do the data support the conclusions?

Reviewer #1: Yes

Reviewer #2: Yes

2. Has the statistical analysis been performed appropriately and rigorously? 

Reviewer #1: Yes

Reviewer #2: Yes

3. Have the authors made all data underlying the findings in their manuscript fully available?

Reviewer #1: Yes

Reviewer #2: Yes

4. Is the manuscript presented in an intelligible fashion and written in standard English?

Reviewer #1: Yes

Reviewer #2: Yes

5. Review Comments to the Author

Reviewer #1: The submitted manuscript titled “Development and validation of a questionnaire assessing household work limitations (HOWL-Q) in women with rheumatoid arthritis” aims to investigate the psychometric properties of a new self-report instrument designed to assess household work limitations in women with rheumatoid arthritis.

The topic is of interest, the study is methodologically sound with potential good empirical contribution.

Here are some aspects that should be addressed:

1. The flow of the Introduction should be improved. For example, lines 87-92 could be moved after line 64.

2. Inclusion/Exclusion criteria were not available.

3. As measures of reliability, other statistics should be used (i.e., McDonald’s omega).

4. Please provide the frequency distribution of the responses to the HOWL-Q individual items using skewness and kurtosis

5. Please specify under which conditions the participants completed the questionnaires in study.

6. In the data analysis, please provide more details regarding psychometric properties (i.e., intern consistency, construct validity) and a better description of the measures used to test convergent validity.

Reviewer #2: The authors proposed a very interesting questionnaire, evaluating household work limitations in rheumatoid arthritis patients. This is an important topic, not condired in the other work addressed questionnaire. Thus, Ii could provide another aspect of work ability impairment in RA women.

The paper is well written and robust from a statistical point of view.

6. PLOS authors have the option to publish the peer review history of their article (what does this mean?). If published, this will include your full peer review and any attached files.

Reviewer #1: No

Reviewer #2: Yes: Fulvia Ceccarelli

---

## [Author Response · Author response to Decision Letter 0]

13 May 2020

RESPONSE TO REVIEWERS AND JOURNAL REQUIREMENTS

• Reviewer #1

The submitted manuscript titled “Development and validation of a questionnaire assessing household work limitations (HOWL-Q) in women with rheumatoid arthritis” aims to investigate the psychometric properties of a new self-report instrument designed to assess household work limitations in women with rheumatoid arthritis.

The topic is of interest, the study is methodologically sound with potential good empirical contribution.

Response: We appreciate the reviewer´s comments. 

Here are some aspects that should be addressed:

1.- The flow of the Introduction should be improved. For example, lines 87-92 could be moved after line 64.

Response: We propose and updated flow for the introduction that includes the reviewer´s suggestion. 

2.- Inclusion/Exclusion criteria were not available.

Response: Inclusion/exclusion criteria had been added to the Methods section.

3.- As measures of reliability, other statistics should be used (i.e., McDonald’s omega).

Response: We have adopted the reviewer´s suggestion and updated the statistical analysis, the results section and reference 32. Thank you for the proposal 

4.- Please provide the frequency distribution of the responses to the HOWL-Q individual items using skewness and kurtosis.

Response: S2 Appendix has been added to the manuscript with the require information, (Results section).

5.- Please specify under which conditions the participants completed the questionnaires in study.

Response: We have added a paragraph in the Procedures section. 

6.- In the data analysis, please provide more details regarding psychometric properties (i.e., intern consistency, construct validity) and a better description of the measures used to test convergent validity.

Response: Internal consistency and construct validity are provided; a description of the questionnaires/index used to test convergent validity is provided in table 1. 

• Reviewer #2

The authors proposed a very interesting questionnaire, evaluating household work limitations in rheumatoid arthritis patients. This is an important topic, not considered in the other work addressed questionnaire. Thus, it could provide another aspect of work ability impairment in RA women.

The paper is well written and robust from a statistical point of view.

Response: We appreciate the reviewer´s comments.

• Journal Requirements:

Response: The PLOS ONE’s style requirements from the manuscript has been reviewed.

2. Please amend your current ethics statement to address the following concerns: Please explain why written consent was not obtained, how you recorded/documented participant consent, and if the ethics committees/IRBs approved this consent procedure.

Response: The ethic statement in the manuscript has been amended. All the participants gave written informed consent. Patients and controls from sample 1 were surveyed to define and rate household tasks/activities; in them, verbal informed consent was approved by local IRB.

(Lertsithichai P. Waiver of consent in clinical observational research. J Med Assoc Thai. 2005;88:275–281)

3. We note you have included a table to which you do not refer in the text of your manuscript. Please ensure that you refer to Table 6 in your text; if accepted, production will need this reference to link the reader to the Table.

Response: We have confirmed that all the tables included have been referred in the text of the manuscript.

---

## [Decision Letter · Decision Letter 1]

1 Jul 2020

Development and validation of a questionnaire assessing household work limitations (HOWL-Q) in women with rheumatoid arthritis.

PONE-D-20-04057R1

Dear Dr. Pascual-Ramos,

We’re pleased to inform you that your manuscript has been judged scientifically suitable for publication and will be formally accepted for publication once it meets all outstanding technical requirements.

Kind regards,

Luca Navarini

Academic Editor

PLOS ONE

Additional Editor Comments (optional):

Reviewers' comments:

Reviewer's Responses to Questions

**Comments to the Author**

1. If the authors have adequately addressed your comments raised in a previous round of review and you feel that this manuscript is now acceptable for publication, you may indicate that here to bypass the “Comments to the Author” section, enter your conflict of interest statement in the “Confidential to Editor” section, and submit your "Accept" recommendation.

Reviewer #1: All comments have been addressed

2. Is the manuscript technically sound, and do the data support the conclusions?

Reviewer #1: Yes

3. Has the statistical analysis been performed appropriately and rigorously? 

Reviewer #1: Yes

4. Have the authors made all data underlying the findings in their manuscript fully available?

Reviewer #1: (No Response)

5. Is the manuscript presented in an intelligible fashion and written in standard English?

Reviewer #1: Yes

6. Review Comments to the Author

Reviewer #1: Thank you very much for the concise and careful revision and clear statements.

Overall, the authors did well in addressing the reviewer's concerns and improving the manuscript.

7. PLOS authors have the option to publish the peer review history of their article (what does this mean?). If published, this will include your full peer review and any attached files.

Reviewer #1: No

---

## [Editor Report · Acceptance letter]

8 Jul 2020

PONE-D-20-04057R1 

Development and validation of a questionnaire assessing household work limitations (HOWL-Q) in women with rheumatoid arthritis. 

Dear Dr. Pascual-Ramos:

I'm pleased to inform you that your manuscript has been deemed suitable for publication in PLOS ONE. Congratulations! Your manuscript is now with our production department. 

Kind regards, 

on behalf of

Dr. Luca Navarini 

Academic Editor

PLOS ONE